# Increasing Confidence in Adversarial Robustness Evaluations

**Roland S. Zimmermann**[*]
University of Tübingen
Tübingen AI Center

**Wieland Brendel**
University of Tübingen
Tübingen AI Center

**Florian Tramèr**
Google

**Nicholas Carlini**
Google

## Abstract

Hundreds of defenses have been proposed to make deep neural networks robust against minimal (adversarial) input perturbations. However, only a handful of these defenses held up their claims because correctly evaluating robustness is extremely challenging: Weak attacks often fail to find adversarial examples even if they unknowingly exist, thereby making a vulnerable network look robust.

In this paper, we propose a test to identify weak attacks and, thus, weak defense evaluations. Our test slightly modifies a neural network to guarantee the existence of an adversarial example for every sample. Consequentially, any correct attack must succeed in breaking this modified network.

For eleven out of thirteen previously-published defenses, the original evaluation of the defense fails our test, while stronger attacks that break these defenses pass it. We hope that attack *unit tests* — such as ours — will be a major component in future robustness evaluations and increase confidence in an empirical field that is currently riddled with skepticism. Online version & code: zimmerrol.github.io/active-tests/

## 1 Introduction

Suppose that someone presents you with a purported proof that P≠NP. The proof is long, complicated, and difficult to follow. How would you go about checking if this proof is correct?

One cumbersome way would be to directly refute the proof's claim, e.g., to demonstrate that actually P=NP by designing an algorithm that solves 3-SAT in polynomial time. While this would definitely refute the proof, it is likely orders of magnitude more difficult than simply showing that the proof is incorrect. Accordingly, researchers typically refute incorrect proofs by studying proofs line-by-line, until they identify some major flaw in the reasoning.

Evaluating adversarial example defenses is somewhat similar to proving a theorem: It is difficult to estimate the robustness of a neural network correctly (just as it is difficult to write down a correct proof), but relatively easy to severely overestimate its robustness (just as it is easy to write down a wrong proof). This contrasts with other areas of machine learning, where performance evaluations are often trivial, e.g., by computing accuracy on some held-out test set. However, evaluating defense robustness necessarily involves reasoning over *all* possible adversaries and showing *none* can succeed. That is, a defense evaluation aims to prove that something is impossible. As a result, despite significant evaluation effort, most published defenses are later broken by stronger attacks [10, 3, 11, 38, 14].

In this light, we here argue for viewing defense proposals as *theorem statements*, and the corresponding evaluations as *proofs*. The purpose of a defense evaluation, then, is to provide a convincing and rigorous argument that the defense is correct. Currently, for an adversary to claim to have a "break" of a defense, it is necessary to actually produce the adversarial examples that cause the model to

---

[*]Work done while at Google. Correspondence to: research@rzrimmermann.com

36th Conference on Neural Information Processing Systems (NeurIPS 2022).

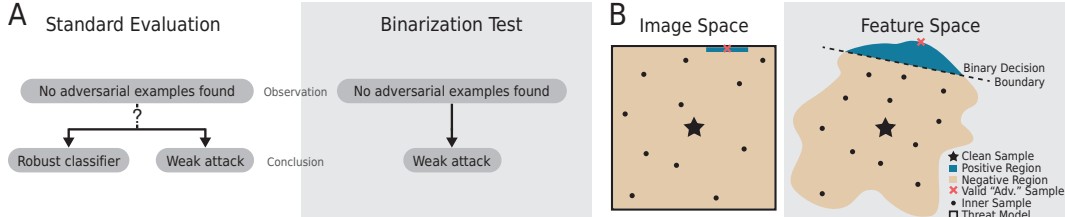

Figure 1: **A: Reasons for apparent high robustness.** There are two reasons an attack might not find an adversarial example. Either the model is robust or the attack is too weak and it could not find the existing adversarials. In our proposed *Binarization Test* we pose a new binary classification problem based on the original classifier such that adversarial examples always exist. Thus, if the attack does not find an adversarial example it follows that the attack is too weak. **B: Setup of the Binarization Test.** We construct a binary classification problem around a clean example such that there exists a valid "adversarial" example within the feasible set of the attack's threat model, here shown for an $\ell_\infty$-bounded attack. Based on the original classifier's features, we create a new binary classifier whose robustness can be evaluated with the same evaluation method as used for the original classifier.

make an error — analogous to refuting a complexity-theoretic impossibility result by producing an efficient algorithm. We argue that this is not how things should work. A valid refutation of a theorem would be to say "there is a flaw in your proof on line 9". This is because the null hypothesis for a theorem is that it is false, just as the null hypothesis for a defense should be that it is not robust.

Unfortunately, for defenses against adversarial examples, outside of studying the actual code used to evaluate (i.e., attack) the defense, there are few opportunities to identify flawed evaluations by reading a research *paper*. Indeed, current best practices for identifying flawed evaluations are limited to looking for artifacts that indicate something has gone (terribly) wrong — for example, that an attack fails even when it is allowed to construct *unbounded* perturbations [11].

We develop a new *active robustness test* to complement existing (passive) tests [11, 26]. Our test designs a new task that is solvable by any sufficiently strong attack. This test can be seen as a "unit test" that the evaluation methodology is correct. Our test purposefully injects adversarial examples into a defense and then checks if the attack used to evaluate the defense is able to find them. If the attack fails this test, we know that the attack is too weak to distinguish between a robust and a non-robust defense, and thus the evaluation should not be trusted.

Returning to our earlier theorem analogy, showing that a defense evaluation fails our test should be viewed similarly as showing a flaw in part of a theorem's proof. Note that this does not necessarily imply that the theorem is not correct (that the defense is not effective), but it should not give us any confidence that it is.

We show that our test would have potentially identified eleven out of thirteen weak evaluations found in peer-reviewed papers. We hope that our testing methodology can become a standard component of future defense evaluations. To this end, defenses with exceptionally novel or different techniques, training algorithms, or architectures, may need to develop their own tailored version of our active unit test, in order to provide convincing evidence that the defense evaluation is indeed correct.

## 2   Background

**Adversarial Examples**   Adversarial examples contain imperceptible perturbations that change the decision of a deep neural network in arbitrary directions [4, 37]. Since they can manipulate the behavior of a model, they are seen as a security concern for machine learning applications. To find adversarial examples for a network, one looks for inputs changing the output of the network while being close (under some norm) to the original data sample. There are a number of methods to solve this optimization problem and to attack a network. Adversarial attacks can be divided into white box methods that use gradient information about the model [e.g., 20, 10, 6, 13, 14], and black box methods that only use the output of the network [e.g., 5, 17, 1, 23, 2].

**Defenses** With an increasing awareness of the risk posed by adversarial examples, a vast number of defenses have been proposed to increase adversarial robustness. For example, some defenses rely on additional input pre-processing steps [e.g., 16], some introduce architectural changes [e.g., 41], and others propose methods for detecting adversarial examples [e.g., 21]. However, most of these defenses eventually turned out to be ineffective against stronger attacks [3, 38]. In terms of general purpose defenses, until now only adversarial training [20] and its variants [e.g., 27, 28, 15] stood the test of time and could not be circumvented. A different approach to defend classifiers against adversarial perturbations are certified defenses which give a theoretical guarantee of the classifier's robustness. However, the robustness of these approaches does not yet reach that of adversarial training [40, 12, 19].

**Challenges in Evaluating Defenses** Properly evaluating the robustness of a model against adversarial examples is non-trivial and there are many potential pitfalls [11]. The critical issue is that when a defense is shown to be robust to a specific attack, this either means that the model is truly robust, or that the attack is suboptimal (see Figure 1A). Possible reasons for an attack to be suboptimal include mechanisms in the model that (often unintentionally) hinder the attack's optimization process or poor choices of hyperparameters [3]. Examples of the former include defenses built around non-continuous activation functions [e.g., 41] or relying on vanishing gradients [e.g., 36]. To address the latter issue, prior work has developed attacks that alleviate the need to manually tune hyperparameters [14]. But as we will show, these attacks are not guaranteed to work well for any model. While the latter issue can be counteracted by using adaptive attacks [38] that are adjusted to a specific model's idiosyncrasies, it remains non-trivial to detect suboptimal attacks in the first place. Previous work suggested guidelines for evaluating the adversarial robustness of a model [11] and developed (passive) indicator values hinting at a failed evaluation [26]. Specifically, Pintor et al. [26] hook various metrics into the execution of *gradient-based* attacks to check for failure cases such as vanishing gradients. Our work goes beyond these passive failure indicators and argues that researchers should *actively* demonstrate that their adversarial attack is sufficiently strong, and that their empirical findings can, thus, be trusted. Our proposed approach also has the advantage of being agnostic to the *type* of attack being used in an evaluation (e.g., gradient-based, decision-based, transfer-based, etc.).

# 3 Active Attack Evaluation Tests

The evaluation of a defense against adversarial attacks becomes more reliable — and the estimated robustness more correct — if the attack is sufficiently strong. The strength of an attack is not an absolute value but depends on the defense it is meant to evaluate, as various defense mechanisms hinder specific attacks [3]. Thus, for a new defense, one needs to demonstrate that the attack proposed to evaluate it is appropriate. In this section, we propose a test that measures the adequacy of a defense's evaluation, and is thereby able to warn researchers of potentially unreliable robustness claims.

To empirically demonstrate the robustness of a defense for some clean input $\mathbf{x}_c$, one runs an attack and shows that it fails to find an adversarial example $\mathbf{x}_{adv}$ within distance $d(\mathbf{x}_c, \mathbf{x}_{adv}) \leq \epsilon$. But can we really be sure there are no adversarial examples in the $\epsilon$ ball if the attack fails? Since the attack cannot guarantee this, there might exist stronger attacks that do find adversarial examples (see Figure 1A).

We propose a test that enables researchers to check whether an attack is too weak to support their robustness claims. In our test, we build a new defense that is as similar as possible to the original, but where we intentionally inject an adversarial example $\mathbf{x}_{adv}$.[2] After building the new (by definition vulnerable) defense, we measure its robustness by running the original evaluation method, and checking whether an adversarial example is found. If the originally used attack fails to find any adversarial example for the modified defense, we should not expect it to properly estimate the robustness of the original defense either.

## 3.1 Test for Classifiers with Linear Classification Readouts

We begin by describing how to construct the modified vulnerable model for a classifier $f$ that consists of a feature extractor $f^*$ followed by a linear classification head. Any standard neural network

---

[2]Intentionally injecting adversarial examples has been (unsuccessfully) considered before as a "honeypot" *defense* [34]. Here, we use this idea in a completely different context, namely to unit-test *attacks*.

architecture falls into this category: the feature extractor $f^*$ comprises every layer except the last, and the linear classification head is the final logit projection layer. Our evaluation methodology keeps the feature extractor $f^*$ unchanged to avoid changing the fundamental behavior of the model, but replaces the classification readout with a newly trained module. This module is trained on a new, specially constructed dataset which allows us to reliably create a classifier where — by design — there exists at least one adversarial example for each sample. A simplified pseudocode implementation of our test is shown in Algorithm 1; all missing functions used there are defined in Appendix A.2.

---

**Algorithm 1** Binarization Test for Classifiers with Linear Classification Readouts. Missing functions are defined in Appendix A.2.

---

**input:** test samples $\mathcal{X}_{\text{test}}$, feature extractor $f^*$ of original classifier, number of inner/boundary samples $N_{\text{i}}$ and $N_{\text{b}}$, distance $\epsilon$, sampling functions for data from the inside/boundary of the $\epsilon$-ball.

**function** BINARIZATIONTEST($f^*, \mathcal{X}_{\text{test}}, N_{\text{b}}, N_{\text{i}}, \epsilon$)
    attack_successful = []
    random_attack_successful = []
    **for all** $\mathbf{x}_c \in \mathcal{X}_{\text{test}}$ **do**
      $h = \text{CreateBinaryClassifier}(f^*, \mathbf{x}_c, \epsilon)$
      # evaluate robustness of binary classifier
      attack_successful.insert $(\text{RunAttack}(h, \mathbf{x}_c))$
      random_attack_successful.insert $(\text{RunRandomAttack}(h, \mathbf{x}_c))$
    ASR = Mean(attack_successful)
    RASR = Mean(random_attack_successful)
    **return** ASR, RASR
**end function**

**function** CREATEBINARYCLASSIFIER($f^*, \mathbf{x_c}$)
    # draw input samples around clean example
    $\mathcal{X}_{\text{i}} = \{\, \mathbf{x_c} \,\} \cup \{\, \text{SampleInnerPoint}(\mathbf{x}_c, \epsilon) \,\}_{1,\ldots,N_{\text{i}}}$
    $\mathcal{X}_{\text{b}} = \{\, \text{SampleBoundaryPoint}(\mathbf{x}_c, \epsilon) \,\}_{1,\ldots,N_{\text{b}}}$
    # get features for images
    $\mathcal{F}_{\text{i}} = \{\, f^*(\mathbf{x}) \mid \mathbf{x} \in \mathcal{X}_{\text{i}} \,\}$
    $\mathcal{F}_{\text{b}} = \{\, f^*(\mathbf{x}) \mid \mathbf{x} \in \mathcal{X}_{\text{b}} \,\}$
    # define labels & create labeled dataset
    $\mathcal{D} = \{\, (\hat{\mathbf{x}}, 0) \mid \hat{\mathbf{x}} \in \mathcal{F}_{\text{i}} \,\} \cup \{\, (\hat{\mathbf{x}}, 1) \mid \hat{\mathbf{x}} \in \mathcal{F}_{\text{b}} \,\}$
    # train linear readout on extracted features
    $g = \text{TrainReadout}(\mathcal{D})$
    $h = g \circ f^*$
    **return** binary classifier $h$ based on feature encoder $f^*$
**end function**

---

In detail, for each test sample $\mathbf{x}_c$ our test consists of the following steps:

**Creating the Dataset**    Initially, we create two collections of input samples which are perturbed versions of $\mathbf{x}_c$

$$\mathcal{X}_{\text{i}} := \{\, \hat{\mathbf{x}} \mid d(\mathbf{x}_{\text{c}}, \hat{\mathbf{x}}) < \xi \cdot \epsilon \wedge \hat{\mathbf{x}} \neq \mathbf{x}_{\text{c}} \,\} \cup \{\, \mathbf{x}_{\text{c}} \,\}_{1,\ldots,N_{\text{i}}} \text{ and } \mathcal{X}_{\text{b}} := \{\, \hat{\mathbf{x}} \mid d(\mathbf{x}_{\text{c}}, \hat{\mathbf{x}}) = \epsilon \,\}_{1,\ldots,N_{\text{b}}}, \quad (1)$$

which are sets of points (randomly sampled) from the inside and the boundary of the $\epsilon$-ball, respectively, with size $N_{\text{i}}, N_{\text{b}} > 0$. Further, $\xi \in (0, 1)$ controls the margin between the inner $\mathcal{X}_{\text{i}}$ and the boundary set $\mathcal{X}_{\text{b}}$. Decreasing $\xi$ effectively increases the gap between inner and boundary points, thus, making it easier to distinguish between the two sets of samples.

Next, for every sample in each of the two sets, obtain the feature representation of the penultimate layer of $f$,

$$\mathcal{F}_{\text{i}} := \{\, f^*(\mathbf{x}) \mid \mathbf{x} \in \mathcal{X}_{\text{i}} \,\} \text{ and } \mathcal{F}_{\text{b}} := \{\, f^*(\mathbf{x}) \mid \mathbf{x} \in \mathcal{X}_{\text{b}} \,\}. \quad (2)$$

**Training the Vulnerable Classifier**    Now, we train a linear (binary) discriminator $g$ that distinguishes samples from $\mathcal{F}_{\text{i}}$ and $\mathcal{F}_{\text{b}}$, i.e., it distinguishes between mildly perturbed images — in the

interior of the $\epsilon$-ball — and some more strongly perturbed images — on the boundary of the $\epsilon$-ball.[3] We want to make sure there exists at least one sample within the threat model's $\epsilon$-ball that $g$ classifies differently than the clean original sample. Due to the construction of the dataset, we can guarantee this by ensuring that $g$ achieves a perfect accuracy on these two sets. If this is not possible for an original sample $\mathbf{x}_c$, we cannot apply the test and, hence, skip the sample[4].

Combining the original classifier's feature extractor $f^*$ with the binary discriminator $g$ yields a new classifier $h = g \circ f^*$ that maps samples to a binary decision. Most importantly, for this new classifier each boundary sample $\mathcal{X}_b$ acts as an $\epsilon$-bounded adversarial example $\mathbf{x}_{\text{adv}}$ for the clean sample $\mathbf{x}_c$.

**Evaluating the Vulnerable Classifier**    We evaluate two properties of this new classifier $h$:

1. The efficacy of the used evaluation method/adversarial attack. For this, one uses the original adversarial attack to attack the modified model $h$ for the clean sample $\mathbf{x}_c$ and records whether an adversarial sample $\mathbf{x}^*$ within the allowed $\epsilon$-ball is found. When calculated and averaged over multiple samples, we call this value the *test score*.
2. The difficulty of the test. To assess this, we use a model-agnostic attack, namely a purely randomized one. We attack the modified classifier $h$ by randomly sampling approximately as many additional data points from within the $\epsilon$-ball around the clean sample $\mathbf{x}_c$ as the adversarial attack queries the model, e.g. for an $N$-step PGD attack [20] use $N$ additional random samples. Finally, one tests whether at least one of them turns out to be an adversarial perturbation for $h$. Averaging over multiple samples yields the *random attack success rate* (R-ASR).

Note that if the classifier $f$ does not use a linear classification readout, one has to modify the test slightly: Instead of using a linear readout for $g$ one needs to use the same type of mechanism that was used originally. While this modification is conceivable for various mechanisms, e.g., k-nearest neighbors classification [35], there might be architectures for which this is harder or impossible, e.g., classification through likelihood estimations based on generative models [32, 45].

## 3.2    Tests for Models Leveraging Detectors

Detection defenses use an additional algorithm that detects and rejects adversarial examples [21]. A successful attack, thus, has to fool both the classifier and the detector, and we have to probe both in our active test.

As earlier, we assume that the classifier can be divided into a feature encoder and linear readout. We make no assumptions on the architecture of the detector, as a wide variety of designs have been proposed in the literature. We define two tests: a *regular* and an *inverted* test. Any reliable evaluation method must pass both. A pseudocode definition of the proposed tests is given as Algorithm 2 in the Appendix.

**Regular Test**    Adversarial examples for a detection defense need to change the classifier's output while also remaining undetected. Thus, to still guarantee that there exists a valid adversarial example, we need to change the construction of the binary classifier slightly to take into account the detector. Specifically, we modify the set $\mathcal{X}_b$ of boundary points such that none of these samples gets rejected by the detector. In practice, we enforce this with rejection sampling, by redrawing boundary points until we find a point that is undetected. Note, that we make no modifications to the detector, since this might require non-trivial optimization of the detector's parameters.

Some adversarial attacks for detector defenses (e.g., feature matching [30]) assume access to reference data samples that belong to a different class but are not adversarial — and, thus, are not rejected. We achieve this by randomly sampling data points outside the $\epsilon$ ball. Thus, we create a new collection

$$\mathcal{X}_r := \left\{ \hat{\mathbf{x}} \mid d(\mathbf{x}_c, \hat{\mathbf{x}}) = \eta\epsilon \right\}_{1,\dots,N_r},$$

---

[3]In some cases, the range of values of the classifier's logits has a large influence on the reliability of an adversarial evaluation (e.g., some weak attacks can fail due to very large or very small logits [10, 3]). To ensure that our modified classifier matches the original classifier as much as possible, we thus aim to mimic the properties of the original classifier's logits. To this end, we rescale the weights of the linear discriminator so as to match the value range of the original classifier's logits.

[4]In our experiments we observe this for only few examples for just the defense of Pang et al. [25]. We attribute this peculiarity to the stochastic nature of the defense.

for which the binary classifier must predict the same class as for the boundary samples $\mathcal{X}_{\mathrm{b}}$. Here, $N_{\mathrm{r}} \geq 0$ and $\eta > 1.0$ control the number of samples and how far outside of the $\epsilon$ ball they are located. Again, as for $\mathcal{X}_{\mathrm{b}}$, we need to ensure that none of these samples get detected. By training the linear readout on $\mathcal{X}_{\mathrm{i}}$, $\mathcal{X}_{\mathrm{b}}$ and $\mathcal{X}_{\mathrm{r}}$ we guarantee that there exists at least one undetected adversarial sample within the $\epsilon$-ball around $\mathbf{x}_{\mathrm{c}}$, and at least $N_{\mathrm{r}}$ samples outside the $\epsilon$-ball that are also undetected.

**Inverted Test**  Since the detector was tuned for a classification problem different from the one posed in the regular test above, it might not work well and rarely reject samples. Thus, a potential issue with the regular test is that a weak attack might pass the test even though the attack completely ignores the detector. Indeed, many evaluations of detector defenses consider weak attacks that are oblivious to the presence of the detector [9]. Thus, an attack passing the test may not be sufficient to tell us that the attack is actually strong enough to successfully target the detector.

To this end, we introduce a second *inverted test* that inverts the attack's goal: Instead of finding adversarial samples that do *not* get detected, the goal is now to find an adversarial example that *does* get detected. Since any detector that claims non-zero robustness must detect some adversarial examples, we use these to construct the set of boundary samples $\mathcal{X}_{\mathrm{b}}$. Finally, we only need to negate the decision of the detection algorithm before proceeding exactly as for the previously described test.

Passing both the regular as well as the inverted test is a necessary condition for an adequate adversarial attack. In fact, this indicates that the attack is not agnostic to the detector but properly takes it into account. In contrast, passing only one of the tests indicates that only the classifier and not the detector is directly targeted.

# 4   Evaluation

We now show that our binarization test could have revealed insufficiently strong evaluations in eleven out of thirteen previously peer-reviewed and published defenses. Of these eleven defenses, nine had already been broken by subsequent attacks — thereby confirming that the original evaluation was indeed too weak. For the other two defenses that fail our test, closer inspection of both defenses' evaluations and code indeed reveal serious issues, and we show that the defenses are broken by stronger attacks. All of these defenses assume an $\ell_{\infty}$ threat model. The specific design choices for the tests adapted to each defense can be found in Appendix B.

**Defenses without Detectors**  We analyze eight defenses which use a classifier with a linear classification readout [8, 41, 43, 22, 25, 33, 31, 44].

We additionally apply our test to two defenses that do not use a simple linear readout to perform classification. However, it is straightforward to adapt the binarization test defined in Algorithm 1 for these classifier architectures. The defense by Verma and Swami [39] leverages an ensemble of readouts. For our test, we therefore also train an ensemble of binary readouts. The classifier of Pang et al. [24] learns to map images to pre-defined class-prototype vectors, and then uses nearest neighbour classification. We reflect this in the test by using two of the class prototypes and associating them with the inner and boundary samples, respectively. Then we re-train a linear layer mapping from features to class prototypes.

In fact, we are the first to show that the defense by Sarkar et al. [31] is less robust than originally reported, as suggested by the fact that it fails our test. All of the above defenses except those by Sarkar et al. [31] were known to be flawed and have been circumvented before.

**Defenses with Detectors**  We investigate three published defenses that aim to detect adversarial perturbations. Following previous work [7], we analyze each defense in a setting where it achieves a false positive rate of $5\%$. While the detection algorithm proposed by Roth et al. [29] runs statistical tests on the classifier's confidence, Shan et al. [34] and Yang et al. [42] analyze earlier activations of the classifier. The first two defenses have been broken before [38, 7] while the latter had not been independently re-evaluated yet.

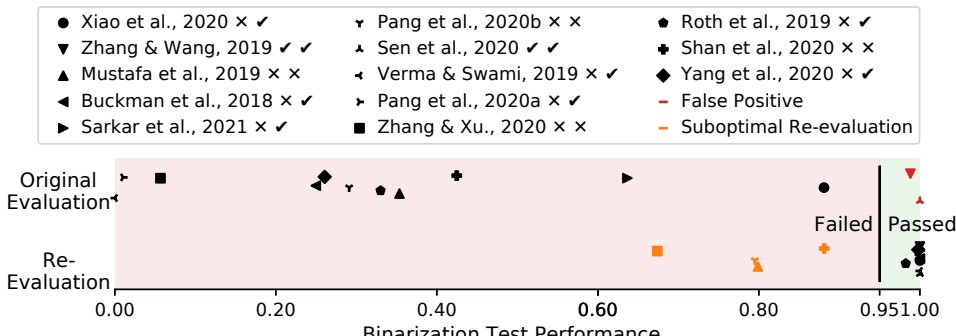

Figure 2: **The binarization test identifies flawed adversarial evaluations.** The x-axis shows the score in our proposed binarization test for the original attack (upper) and a subsequent improved attack (lower). We define a threshold of $0.95$ that attacks need to achieve to pass our test. For detector-based defenses, we visualize the minimum of the performance on the regular and inverted tests. Note that for each defense, the subsequent improved attack substantially decreases the defense's robust accuracy (by at least 12%). Black markers indicate original attacks that fail the test, as well as improved attacks that pass the test (i.e., true positives and true negatives for our test). Red markers indicate suboptimal original evaluations that nevertheless pass our test (false positives). Orange markers indicate re-evaluations that used suboptimal attacks (as shown by our test) that still broke the defense. We discuss these cases in Section 4.2. Checks and crosses in the legend indicate passing/failing tests for the original and the re-evaluation, respectively. See Figure 4 in the Appendix for the robust accuracies.

## 4.1 Initial Evaluation of Not-previously-broken Defenses

We begin by investigating the two recent and not yet broken defenses. Here, we are interested in seeing whether the original robustness evaluations pass our binarization test. While a positive result would increase confidence in the defenses' claims, a negative outcome would cast doubts.

**Sarkar et al. [31]** The original evaluation of this defense fails our test with a test score of $0.04$. That is, the original attack only finds an adversarial example in the modified binary classifier 4% of the time — even though at least one adversarial example is guaranteed to exist for every test sample. This is strong evidence that the attack is weak and thus the robustness claim likely overestimated. Upon investigation, we found a flaw in the original evaluation's code: The statistics of the batch normalization layers are not frozen during evaluation, which changes the behavior of the model during the attack. Properly freezing these layers at inference and increasing the number of PGD steps from 20 to 75 yields a perfect score $(1.0)$ in our binarization test. Moreover, this updated evaluation methodology reduces the robust accuracy to $\leq 1\,\%$ down from the originally reported $60.15\,\%$ and, thus, effectively breaks the defense.

**Yang et al. [42]** For this detector-based defense, we find that the attack used in the original evaluation is agnostic to the detector and only targets the classifier. Consequentially, this attack fails both the regular and the inverted binarization test with a low score of $0.26$ and $0.63$, respectively. We thus create a new adversarial attack based on PGD that combines two objectives: (1) fool the classifier by maximizing the adversarial loss and (2) stay undetected by matching the features of a non-adversarial sample as much as possible (a feature matching attack [30, 38]). This adaptive attack achieves a nearly perfect score of $0.99$ in the test and reduces the robust accuracy of the defense from the originally reported $99\,\%$ down to below $12\,\%$.

## 4.2 Interpreting Test Results for Weak and Strong Attacks

Since eleven of the considered defenses have already been broken before, and we showed how to break the remaining two, we now have access to both a flawed and a well-working adversarial evaluation method for each defense. This allows us to compare how these attacks perform in terms of both the estimated robust accuracy and the score on the binarization test. We visualize the results in Figure 2. For eleven out of the thirteen considered defenses, our proposed test would have flagged

their evaluation as insufficient: the original attacks' test performance is substantially below a perfect score (i.e., $< 0.95$). Furthermore, the test scores improve for all defenses when replacing the original evaluation code with an improved attack (the defense of Sen et al. [33] is an exception, as the original evaluation already obtains a perfect test score due to an integration bug, as described below).

**Explaining the False Positives**    Our test incorrectly lets two defense evaluations that had bugs pass (see red markers in Figure 2). When investigating these failure cases in more detail, we find that the original attack used by Sen et al. [33] is not flawed or incorrectly implemented per se, but it is not used correctly. Namely, the attack generates adversarial examples with respect to the classifier's predicted label, instead of the ground-truth label. As a result, for some misclassified samples the attack actually *corrects* the classifier's mistake! By switching to an attack that correctly targets the ground-truth label, we reduce the robust accuracy drastically.

Our test is unable to catch such a mistake: By design, our test constructs a binary classifier with 100% accuracy (and thus the classifier's predicted label is always equal to the ground-truth). If we view our proposed test as a *unit test* for an attack, then the type of bug in the above evaluation is akin to an *integration bug*, where the (correct) attack is called with incorrect parameters.

For the defense by Zhang and Wang [43] we notice, a high R-ASR value ($> 0.75$) that we could not decrease further. Thus, for this defense, our binarized classifier is too easy to attack. We hypothesize that by increasing the number of inner samples $N_i$ substantially, the test might become hard enough to indicate sub-optimal evaluations for this defense.

**Explaining the Suboptimal Re-evaluations**    There are also four defenses for which the improved attacks still fail our test, even though their test performance is better than for the original attacks. The authors of the improved attack for Pang et al. [25] note that while this attack already breaks the defense, one could improve the attack further [38]. For the defenses by Mustafa et al. [22] and Zhang and Xu [44], the improved attacks are not adaptive attacks but part of AutoAttack's attack collection [14]. Although these attacks were sufficient to drastically reduce the measured robustness of the defenses (see Appendix, Figure 4), they are likely not optimal attacks for these defenses. While the attack [7] used for the re-evaluation breaks the defense by Shan et al. [34], the imperfect test score hints at the possibility of an even more potent and yet-to-be-discovered attack.

### 4.3   Hardness of the Test

To put the performance that an adversarial attack achieves in the binarization test into perspective, we quantify the hardness of the test using the previously introduced random attack success rate (R-ASR). Comparing it to the test result of the attack allows us to deduce how effective the attack is in finding adversarial examples for the model in question.

There are several parameters and design choices relevant for our test that influence its hardness. For one, by increasing $N_i$ we train the binary discriminator on a larger number of different non-adversarial points which increases the robustness of the discriminator and, thus, makes the test harder. Conversely, by increasing $N_b$ we plant a larger number of adversarial examples for the discriminator within the $\epsilon$-ball, making the test simpler.

Even with a large but finite number of training samples, there is no unique solution for the binary discriminator but instead a set of valid solutions. While all of these classifiers have perfect accuracy on the training set, they differ in how close the decision boundary is placed to the boundary samples. The closer the decision boundary is placed to the boundary samples, the smaller the volume of valid adversarial examples and, thus, the harder the test becomes. The effect of the the decision boundary's closeness on the test's hardness is illustrated in Figure 3 for the defense of Mustafa et al. [22]. Here, placing the boundary closer to the boundary samples decreases both the R-ASR as well as the ASR of two sub-optimal attacks while that of a better suited attack stays robustly at $1.0$.

On the one hand, a test that is too easy has no predictive power about the attack's efficacy (since any attack might trivially pass it), while on the other hand, a test that is too challenging might actually underestimate the attack's true performance (i.e., finding an adversarial example in the modified defense might be harder than in the original defense, and thus even a strong attack against the defense might fail the test if it is tuned too aggressively). Therefore, one needs to tune the test's hardness to a reasonable level.

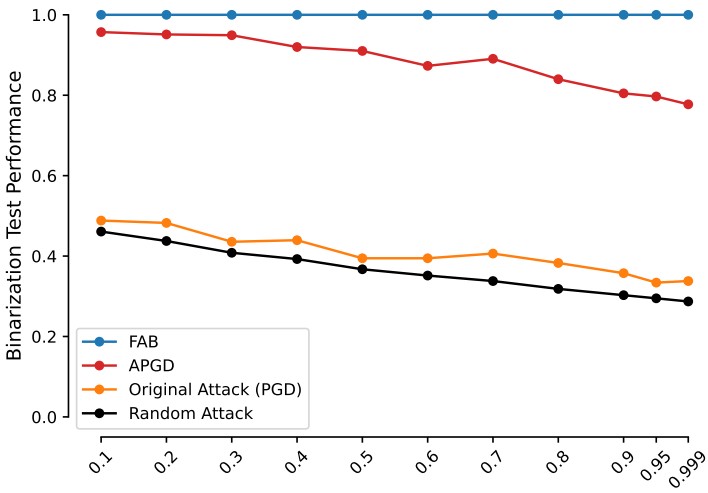

Test Hardness Measured by Closeness $\kappa$ of Decision Boundary

Figure 3: **Hyperparameters influence the test's hardness.** For the defense by Mustafa et al. [22], we compare the test performance of three attacks: two sub-optimal attacks, namely the originally used PGD attack (orange) and AutoPGD (red) [14], yielding robust accuracies of $32.32\,\%$ and $8.16\,\%$ and the stronger FAB attack (blue) [13] yielding $0.71\,\%$. As one indicator of the test's hardness, we show the ASR of a random attacker (R-ASR, black). The test's hardness is controlled by the hyperparameter $\kappa$ which, in feature space, measures the distance between the decision boundary and the boundary sample relative to the distance between boundary and the closest inner sample. Note that the larger $\kappa$ is, the closer the decision boundary is to the boundary sample.

We recommend the following procedure to adjust the test's hardness: To ensure we do not overestimate the test performance (since this is the more dangerous direction), we start with a configuration that makes the test as hard as possible. Then, decrease the hardness until the adversarial attack in question reaches an (almost) perfect ASR. Note that if there is no configuration that yields a near-perfect ASR, then the attack did not pass the test and one should be skeptical of the attack's ability to properly estimate the classifier's robustness. Finally, compare the ASR with the R-ASR (the success rate of a random attack): If the ASR is not substantially higher — or is even lower — than the R-ASR, this is strong evidence that the attack performs poorly. If instead the gap is large, the attack has passed this necessary test and might be powerful enough to properly estimate the classifier's robustness.

## 5 Discussion & Conclusion

This paper made a case for *active* tests to evaluate adversarial robustness. The goal of an active test is to provide compelling evidence that an attack has sufficient power to evaluate a classifier's robustness. We presented such a test for defenses using linear classification readouts and showed how to adapt this test for different defense mechanisms such as detector-based defenses. The type of test proposed in this work acts as a necessary condition for robustness evaluations, i.e., an attack that fails the test will most likely overestimate the defense's robustness.

While we have presented a potential test that could help defense authors demonstrate sufficient power of their adversarial evaluation, our tests are not meant to be comprehensive and directly applicable to every possible defense. For example, our tests are primarily designed to work for defenses that use linear classification readout layers. If a defense were to have a different classification layer instead, such as a k-Nearest Neighbor classifier, then our tests would need to be modified accordingly. Consequentially, defense authors should aim to develop their own attack unit tests, depending on the particular claims made.

Our test could have revealed weak evaluations in eleven out of thirteen previously peer-reviewed (and subsequently broken) defenses. We are thus optimistic that active tests can improve the reliability of future publications in the field of adversarial robustness.

## Acknowledgements

We thank Alexey Kurakin, Evgenia Rusak, Prasanna Mayilvahanan, Thaddäus Wiedemer and Thomas Klein for their valuable feedback. The authors thank the International Max Planck Research School for Intelligent Systems (IMPRS-IS) for supporting RSZ. This work was supported by the German Federal Ministry of Education and Research (BMBF): Tübingen AI Center, FKZ: 01IS18039A. WB acknowledges financial support via an Emmy Noether Grant funded by the German Research Foundation (DFG) under grant no. BR 6382/1-1 and the Open Philanthropy Foundation funded by the Good Ventures Foundation. WB is a member of the Machine Learning Cluster of Excellence, EXC number 2064/1 – Project number 390727645.

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
