# A Algorithms

## A.1 Binarization Test for Detection Defenses

To expand the test outlined in Algorithm 1 to detection defenses, we propose the following modification:

---

**Algorithm 2** Binarization Test for classifiers with a linear classification readout and a detector. Missing functions are defined in Appendix A.2.

---

**input:** test samples $\mathcal{X}_{\text{test}}$, feature extractor $f^*$ of original classifier, adversarial detector $d$ returning $1$ for detected samples and $0$ otherwise, number of inner/boundary/reference samples $N_{\text{i}}/N_{\text{b}}/N_{\text{r}}$, distance $\epsilon$, sampling functions for data from the inside/boundary of the $\epsilon$-ball, relative distance (in terms of $\epsilon$) of positive and reference samples $\eta > 1$.

**function** BINARIZATIONTEST($f^*, d, \mathcal{X}_{\text{test}}, N_{\text{b}}, N_{\text{i}}, N_{\text{r}}, \epsilon, \eta$)
    attack_successful $= []$
    random_attack_successful $= []$
    **for all** $\mathbf{x}_c \in \mathcal{X}_{\text{test}}$ **do**
        $b, \mathcal{X}_{\text{r}} =$ CreateBinaryClassifier($f^*, \mathbf{x}_c, \epsilon$)
        # evaluate robustness of binary classifier
        attack_success.insert(RunDetectorAttack($b, d, \mathbf{x}_c, \mathcal{X}_{\text{r}}$))
        random_attack_success.insert(RunRandomDetectorAttack($b, d, \mathbf{x}_c$))
    ASR $=$ Mean(attack_successful)
    RASR $=$ Mean(random_attack_successful)
    **return** ASR, RASR
**end function**

**function** INVERTEDBINARIZATIONTEST($f^*, d, \mathcal{X}_{\text{test}}, N_{\text{b}}, N_{\text{i}}, N_{\text{r}}, \epsilon, \eta$)
    # $\neg d$ denotes the negated/inverted detector
    **return** BinarizationTest($f^*, \neg d, \mathcal{X}_{\text{test}}, N_{\text{p}}, N_{\text{n}}, \epsilon, \eta$
**end function**

**function** CREATEBINARYCLASSIFIER($f^*, \mathbf{x_c}, d$)
    # draw input samples around clean example
    $\mathcal{X}_{\text{i}} = \{ \mathbf{x_c} \} \cup \{ \text{SampleInnerPoint}(\mathbf{x}_c, \epsilon) \}_{1,\ldots,N_{\text{i}}}$
    $\mathcal{X}_{\text{b}} = \{ \text{SampleBoundaryPoint}(\mathbf{x}_c, \epsilon), d(z) = 1 \}_{1,\ldots,N_{\text{b}}}$
    # get positive samples outside the $\epsilon$-ball, e.g., as a reference for logit matching attacks
    $\mathcal{X}_{\text{r}} = \{ \text{SampleBoundaryPoint}(\mathbf{x}_c, \eta\epsilon), d(z) = 1 \}_{1,\ldots,N_{\text{r}}}$
    # get features for images
    $\mathcal{F}_{\text{i}} = \{ f^*(\mathbf{x}) \mid \mathbf{x} \in \mathcal{X}_{\text{i}} \}$
    $\mathcal{F}_{\text{b}} = \{ f^*(\mathbf{x}) \mid \mathbf{x} \in \mathcal{X}_{\text{b}} \}$
    $\mathcal{F}_{\text{r}} = \{ f^*(\mathbf{x}) \mid \mathbf{x} \in \mathcal{X}_{\text{r}} \}$
    # define labels & create labeled dataset
    $\mathcal{D} = \{ (\hat{\mathbf{x}}, 0) \mid \hat{\mathbf{x}} \in \mathcal{F}_{\text{i}} \} \cup \{ (\hat{\mathbf{x}}, 1) \mid \hat{\mathbf{x}} \in \mathcal{F}_{\text{b}} \} \cup \{ (\hat{\mathbf{x}}, 1) \mid \hat{\mathbf{x}} \in \mathcal{F}_{\text{r}} \}$
    # train linear readout on extracted features
    $b =$ TrainReadout($\mathcal{D}$)
    **return** binary classifier $b$ based on feature encoder $f^*$ and reference samples $\mathcal{X}_{\text{r}}$
**end function**

---

## A.2 Auxiliary Functions

In order to keep the main algorithms concise, we use the following auxiliary functions:

**TrainReadout($\mathcal{D}$)** Trains a binary classification readout on the labeled dataset $\mathcal{D}$ that uses the same architecture as the original classification readout of the defense in question.

**RunAttack($h, \mathbf{x_c}$)** Searches for an adversarial example for the clean example $\mathrm{x}_c$ that fools the binary classifier $h$ using the adversarial attack in question.

**RunRandomAttack($h, d, \mathbf{x_c}$)** Searches for an adversarial example for the clean example $\mathbf{x_c}$ that fools the binary classifier $h$ using a purely random attack, i.e., $L$ random perturbations are drawn and checked whether at least one of them is adversarial. In our experiments we sample, we set $L = 400$ (see Section B).

**RunDetectorAttack($b, d, \mathbf{x_c}, \mathcal{X_r}$)** Searches for an adversarial example for the clean example $\mathbf{x_c}$ that fools the binary classifier $h$ and the detector $d$ using the adversarial attack in question. The set $\mathcal{X_r}$ contains dataset samples of different classes that might be used by feature matching attacks.

**RunRandomDetectorAttack($b, d, \mathbf{x_c}$)** Searches for an adversarial example for the clean example $\mathbf{x_c}$ that fools the binary classifier $h$ and the detector $d$ using purely random attack, i.e., $L$ random perturbations are drawn and checked whether at least one of them is adversarial. In our experiments we sample, we set $L = 400$ (see Section B).

**SampleInnerPoint($\mathbf{x_c}, \epsilon$)** Draws a random sample from inside of the $\epsilon$-ball, i.e., $0 < d(\mathbf{x_c} < \epsilon$ for $d$ being the distance measure. In our experiments we sample from a uniform distribution (see Section B for more details).

**SampleBoundaryPoint($\mathbf{x_c}, \epsilon$)** Draws a random sample that lies exactly on the surface of the $\epsilon$-ball, i.e., $d(\mathbf{x_c} = \epsilon$ for $d$ being the distance measure. In our experiments we sample from a uniform distribution (see Section B for more details).

# B  Experimental Details

For all defenses considered, we use the source code and hyperparameters originally used by the defenses' authors to evaluate them. For integrating our test in the respective evaluations, we aimed to minimally modify the original code.

The experiments were run on a computer with eight NVIDIA GeForce RTX 2080 Ti GPUs, whereby for each experiment only a single GPU was used. In total, approximately 2500 GPU hours were required for obtaining the results presented in this work.

All investigated defenses consider an $\ell_\infty$ threat model. While the defense by Shan et al. [34] focuses on an $\epsilon = 0.01$ bound, the rest uses the more common $\epsilon = 8/255$ bound.

We evaluate the binarization test for 512 randomly chosen samples from the CIFAR-10 [18] test set.

For all attacks we set the gap between the boundary and inner points to $\eta = 0.05$, measured relatively to the used $\epsilon$ value. We evaluated detector-based defenses using Algorithm 2, and use $\xi = 1.75$, measured in terms of $\epsilon$.

As outlined above in Section 4.3, we adjust the hardness of the test until the test produces conclusive results, i.e., the random attack success rate (R-ASR) is not too high. This leads to a parameter choice of $N_{\text{inner}} = 999$ for all defenses but that of Zhang and Wang [43] for which used $N_{\text{inner}} = 9999$. While we set the number of boundary samples to $N_{\text{boundary}} = 10$ for Zhang and Xu [44], we set it to 1 for all other defenses. Also, we sample the boundary point(s) from the corners of the $\ell_\infty$ $\epsilon$-box, since this increases the test's difficulty further.

Further, for adjusting the hardness of the test we adjust the bias of the linear classifier such that the distance between boundary sample and decision boundary measured in terms of the distance between boundary sample and closest inner sample is $\kappa = 0.999$ (see Section 4.3).

We sample the inner samples uniformly from the $\epsilon$ hypercube (i.e., the $\ell_\infty$ ball), and the boundary samples from the corners of the cube. We opted for this, since it increases the hardness of the test. Further, for calculating the R-ASR we samples both 200 points from the inner and 200 more from the corners of the space, as this significantly increased the R-ASR and, thus, gives a more realistic estimate of the test's difficulty.

# C  Additional Results

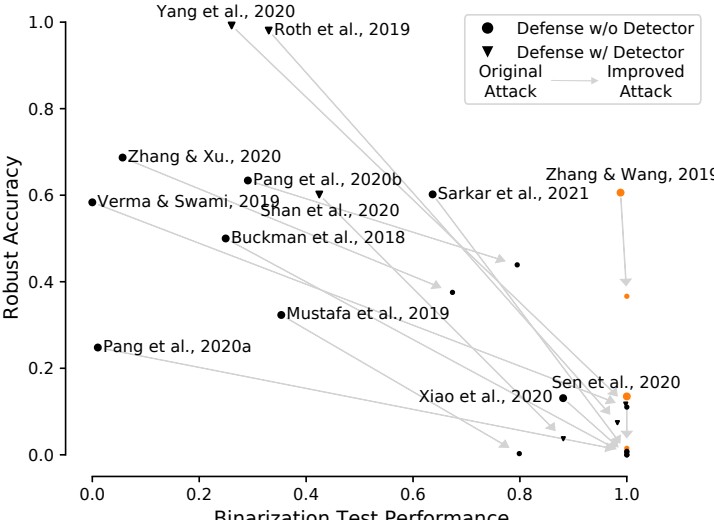

Figure 4: **Robust accuracy as a function of the test performance.** Thicker markers denote results for the attacks originally used by the defenses' authors, while smaller ones correspond to that of adaptive attacks that broke the defense. The gray arrows between these points indicate how the scores change by using using a better suited attack. Orange points indicate false negatives/non-conclusive test results. Triangles denote defenses leveraging detection algorithms. For these defenses we visualize the minimum of the performance on the regular and inverted tests.