# OpenReview forum: "Increasing Confidence in Adversarial Robustness Evaluations"
_NeurIPS.cc/2022/Conference — NeurIPS 2022 Accept_

### Official Review · Reviewer_KmTn · 2022-07-07

**Rating:** 8
**Confidence:** 5
**Soundness:** 4 excellent
**Presentation:** 4 excellent
**Contribution:** 4 excellent

**Summary:**

This paper proposes a binarization test to identify weak attacks against adversarial example defenses. The proposed test changes the model’s prediction layer to a binary classifier and fine-tunes it on a small crafted dataset for each benign example. As a result, the original attack, if sufficiently strong, should be able to find adversarial examples when applied to the modified model. This test serves as an active robustness test to complement existing passive tests of weak attacks. Empirical results show that the proposed test effectively re-confirmes the weak evaluations of 11/13 previous defenses, and two of them were not discovered before.

**Questions:**

I would recommend that the authors elaborate on the four weaknesses in the Quality section.

**Limitations:**

The lack of evaluation of adversarial training might be a potential weakness, but the existing evaluation is already sufficient and representative.

**Strengths And Weaknesses:**

### Originality

**Strengths (major)**
* The proposed active test provides a novel perspective for identifying weak evaluations of adversarial examples defenses.
* The overall idea and approach are novel and insightful.
* The design of evaluating the difficulty of the proposed test is good, as it provides validation when a weak attack passes the test.

**Weaknesses (minor)**
* **Unclear algorithmic improvements to [34].** The authors mentioned in footnote 1 (page 3) that a similar idea was used before as a honeypot defense [34]. It is suggested to discuss how the algorithm (e.g., injecting adversarial examples) used in this paper differs from those used in [34]. For example, are there any new challenges when directly adopting the previous approach, are there any insightful modifications to that approach so it fits the proposed test, or does the idea of [34] fit more in detecting weak attacks rather than as a defense?
* **Lower bound for Equation (1).** Does Equation (1) need a lower bound for the inner point sampling to guarantee the minimum hardness of the test?

### Quality

**Strengths (major)**
* The evaluation is strong and clearly demonstrates the effectiveness of the proposed active test.
* Two previous weak evaluations are discovered.

**Weaknesses (minor)**
* **Unclear effect of changing the classification head.** While it makes sense as a test to change the prediction head, I am not sure if that would negatively affect the test as the changed model may not precisely “mimic” the original model. For example, if we were to evaluate adversarial training, the fine-tuned prediction head may lead to a less robust model than the original one. As a result, it seems easier to attack the test model than the original one.
* **Applicability to adversarial training.** While I understand that evaluating skeptical defenses is more straightforward in demonstrating the effectiveness of the proposed test, it is suggested to discuss the proposed test’s applicability to adversarial training. In particular, I am curious if the test could detect weak attacks (e.g., weak PGD with a large step size or a few steps) on adversarially trained models (with different robustness). This might be informative as the robustness from adversarial training is also developing, and weak attacks (PGD vs. AutoPGD) may overestimate its robustness.
* **Unclear overheads.** It is suggested to include the overheads of Algorithm 1, as it trains a new prediction head for each test sample. It is also unclear how many test samples are needed to produce a confident claim of passing the test. At L138, the authors mentioned that some samples might not produce a model with perfect accuracy. I am curious why this would happen and if that affects the overheads significantly.
* **Slightly confusing clarification of false positives.** The clarification at L261-263 is slightly confusing, as I did not expect misclassified samples to be included in the evaluation from the beginning. If included, it may not be the attack that "is not used correctly," but the evaluation (of this paper) should follow the same setting as the integrated attack. This further leads to the confusion of why the label matters here: If all points are sampled without labels in Equation 1, and the RunAttack in algorithm 1 refers to the exact original attack with the exact choice of label, then it seems that we should not observe the artifact at L265-268.

### Clarity

**Strengths (major)**
* The paper is generally well-written.

**Weaknesses (misc)**
* The legend for the threat model in Figure 1 is confusing; maybe it should be an empty square.
* I feel that some contributions can be highlighted earlier in the paper. For example, the inclusion of detection defenses and the newly identified flaws.
* Most functions in Algorithm 1 are not clearly defined and may need some clarification or reference to corresponding texts. For example, RunAttack, RunRandomAttack, SampleInnerPoint, TrainReadout, etc.
* The statement at L190-192 is hard to follow and needs more motivation and clarification.

### Significance

**This paper provides strong positive results for more confident evaluations of adversarial example defenses. While I have listed several weaknesses above, most are not major and are suggested as discussions that further strengthen the paper.**

---

> ### Author Response · Authors · 2022-08-02
> **Response to Reviewer KmTn 1/2**
>
> Dear Reviewer,
>
> Thank you very much for your positive review and your valuable feedback! We are very happy that you perceived our work as “novel”, “insightful” and “well-written” with a “strong evaluation”.
>
> Please find our responses to your points below:
>
> **Comment:** _“How [does] the algorithm used in this paper differs from those used in [34]?”_ \
> **Answer:** Thanks for asking this question. We see that the footnote can be misleading. What we meant to say is that there is a weak similarity between [34] and our work, in the sense that both methods use the idea of injecting adversarial examples into a classifier. However, this is where the similarity ends. Our work and [34] differ in both motivation (theirs is an adversarial example defense while ours is a test to identify weak defense evaluations) and methodology.
>
> **Comment:** _“Does Equation (1) need a lower bound for the inner point sampling?”_ \
> **Answer:** Yes, you are correct, that the number of inner samples affects the hardness of the problem (see Section 4.3, lines 287-291). We are not aware of a theoretical lower bound on the number of inner samples, however, empirically we see that this number needs to be set high compared to the number of boundary samples (e.g. 999 vs 1 in most of our experiments, see Appendix B).
>
> **Comment:** _“Unclear effect of changing the classification head.”_ \
> **Answer:** You are correct that changing the classification head might change the efficiency of adversarial attacks against the model (e.g., if the classification readout performs gradient masking due to enormous logits). To ease this issue, we aim to “mimic” the original classification head as much as possible with the replacement, e.g. reproduce similar logit values. Further, we are not aware of any known defense that explicitly increases robustness with its classification readout: For example, for adversarial training, it was shown by Engstrom et al. 2019 [1] that robustness rather comes from robust features than from a special readout mechanism.
>
> **Comment:** _“[Could] the test [...] detect weak attacks [...] on adversarially trained models?”_ \
> **Answer:** Thanks for raising this interesting point. To investigate this, we applied our test to different variants of PGD (different step sizes, number of steps) that attacked an adversarially trained ResNet50 [2]. Our preliminary results suggest that PGD attacks with too few steps (e.g., if $\textrm{step}_\textrm{size} \cdot n_\textrm{steps} < \epsilon$) for which the model appear robust, result in low test performance, i.e. our test can identify these weak evaluations. We will investigate this further and add a full ablation study on this in the camera-ready version of the paper.
>
> **Comment:** _“Unclear overheads.”_ \
> **Answer:** Thank you for raising this question. As described in the text, for each clean data sample, we need to run three steps which each contribute to the computational cost: First, we obtain the features of $N_{inner}$ + $N_{outer}$ samples (i.e. performing $(N_\textrm{inner}+N_\textrm{outer}) / \textrm{batch size}$ forward passes). Second, we train a binary classifier on these features, e.g., using logistic regression which, in our experiments, was computationally neglectable. Finally, we run the attack for a single sample - depending on the attack this is the most costly step as this also cannot be parallelized for multiple clean data samples in question as the weights of the classification layer differ for each of them. Thus, if we want to run our test on $N$ samples, its effective computational cost is mostly dominated by the cost of running the attack $N$ times.
>
> **Comment:** _“Slightly confusing clarification of false positives.”_ \
> **Answer:** It is correct that no label information is used in sampling the data points we use to train the binary readout. We believe that there is a misunderstanding on what caused the false positive result in this case: The attack used by the defense’s authors is strong and reasonable - the authors of the defense just had a bug in their call to the attack during the evaluation, which is only relevant if the model makes mistakes for clean samples. However, since we run our test only for clean samples for which the newly trained binary classifier module got a perfect score, none of these cases exist.
>
> **Comment:** _“Figure 1[b] is confusing.”_ \
> **Answer:** We are sorry that this figure was not as clear as we hoped it would be. We updated and expanded the caption of Figure 1B to make it easier to parse the figure (i.e., we explained that the black box depicts the feasible set of an l-infinity norm-bounded attack).
>
> **Comment:** _“functions in Algorithm 1 are not clearly defined and may need some clarification or reference to corresponding texts.”_ \
> **Answer:** Thanks for pointing this out. We now ensured that all external methods used in the algorithms are properly described (see Appendix A of the updated version).

---

> ### Author Response · Authors · 2022-08-02
> **Response to Reviewer KmTn 2/2**
>
> **Comment:** _“The statement [in] L190-192 is hard to follow.”_ \
> **Answer:** As outlined in L184-189, many detection defenses are actually (wrongfully) evaluated by attacks that are oblivious to the detector. This creates a false sense of security, as an attack that is not oblivious to the detector might still break the defense. In a nutshell, the “inverted test” just ensures that the attack in question is capable of attacking both the detector and classifier. For this, we introduce a new detector (that is the negated version of the original detector) and check if the attack can still find adversarials for this new detector. We will update this description in the camera-ready version of our paper.
>
> [1] Engstrom, L., Ilyas, A., Santurkar, S., Tsipras, D., Tran, B., & Madry, A. (2019). Adversarial robustness as a prior for learned representations. arXiv preprint arXiv:1906.00945.
>
> [2] Engstrom, L., Ilyas, A., Salman, H., Santurkar, S. and Tsipras, D.. Robustness (Python Library). 2019.

---

> > ### Comment · Reviewer_KmTn · 2022-08-05
> > **Thank you for your detailed response!**
> >
> > I appreciate the detailed response, and my remaining concerns are clarified. I would recommend adding as many as these clarifications to the final version.
> >
> > A minor follow-up on the legends of Figure 1b, as I see that other reviewers also mentioned this -- What confused me was that the threat model's legend is ■, yet I could not find ■ in the figure, even if I know that it refers to the square boundary. Hence, I thought using □ might be more clear, as this is what the boundary appears in the figure.

---

> > > ### Author Response · Authors · 2022-08-09
> > > **Reponse to Reviewer KmTn**
> > >
> > > Thank you for this explanation and making us aware of this! We know see how this part of the legend can be confusing and will update the legend accordingly for the final version of the paper.

---

### Official Review · Reviewer_Wz6X · 2022-07-09

**Rating:** 6
**Confidence:** 4
**Soundness:** 3 good
**Presentation:** 2 fair
**Contribution:** 3 good

**Summary:**

Hundreds of adversarial attacks and defenses have been proposed in the last few years. How Hundreds of adversarial attacks and defenses have been proposed in the last few years. How robust defense depends on the choice of evaluation. All defenses can be broken given enough perturbation. This paper presents a model, defense, and attack agnostic methodology to identify weak defenses. The authors propose a binarization test. (1) Create a 2-class synthetic dataset based on real examples which are in-boundary and on the boundary
 (2) Take the feature extractor of a robust classifier (f*) and train a binary classifier (g) that can classify the dataset (3) Evaluate the attack on the h = (g o f*) classifier and compute the % of times the attack is successful. Higher this score, more potent the attack. 11/13 previously published defenses failed this test, showing that the defenses are not strong but the evaluation is weak.


**Questions:**

- How many samples are you using in training the classifier g? Do you have any ablation on this?
- Do you have any ablation on the strength of the attack $\epsilon$ and how the attacks fare against the binarization test?

**Strengths And Weaknesses:**

Strengths: The authors address an important issue of robustness evaluation in the adversarial literature. The binarization test technique is novel and it is both simple and computationally cheap. The authors discuss the failure case, I appreciate it.

Weakness: The writing is very unclear. The idea is simple, but it is really hard to understand from the writing. Things like figures and even algorithms that are supposed to convey the message easily are hard to parse. It needs a major re-write to simplify things if it were to be accepted into the conference.

---

> ### Author Response · Authors · 2022-08-02
> **Response to Reviewer Wz6X**
>
> Dear Reviewer,
>
> We thank you for your positive review and helpful feedback! It is encouraging to see that you acknowledge our work addresses an important issue using a “novel”, “simple” and “computational cheap” method.
>
> Please find our responses to your questions and comments below:
>
> **Comment:** _“writing is very unclear”, “figures and even algorithms are hard to parse”_ \
> **Answer:** We updated the description of Figure 1 and expanded the description of the algorithms to make them easier to parse. Furthermore, we ensured that all methods used in the algorithms are defined (see Appendix A of the updated version).
> Regarding the writing, we are unsure what parts or aspects you found unclear or confusing. We would be grateful for any specific pointers and will be happy to incorporate your suggestions to increase the manuscript's clarity.
>
> **Comment:** _“How many samples are you using in training the classifier g?”_ \
> **Answer:** We train the binary classification readout on the features of $N_\textrm{inner} + N_\textrm{boundary}$ samples. In our experiments, this amounts to 1000 samples for all defenses except that by Zhang et al. 2020 (10000) and Zhang et al. 2019 (1009). For a more detailed description of the chosen parameters, please refer to Appendix B (line 593 of the original and 557 of the revised manuscript).
>
> **Comment:** _“Do you have any ablation on the strength of the attack ϵ?”_ \
> **Answer:** Since we wanted to stay as close as possible to the original evaluation of the defenses investigated in our study, we used their attack settings. As most defenses only report these for a single ϵ, we could not run a large-scale ablation on the influence of ϵ on the test.

---

> > ### Comment · Reviewer_Wz6X · 2022-08-08
> > **Thank you for the response.**
> >
> > Thank you for answering my minor queries. I would like to maintain my positive score at this point.

---

> > > ### Author Response · Authors · 2022-08-09
> > > **Response to Reviewer Wx6X**
> > >
> > > Dear Reviewer,
> > >
> > > Thank you for your response and your overall positive assessment of our work! We would be grateful if you could let us know what aspects of the paper would need to be improved so you would consider a higher overall assessment. We will do what we can to address any remaining concerns and thank you again for your constructive review.

---

### Official Review · Reviewer_Rijw · 2022-07-11

**Rating:** 6
**Confidence:** 4
**Soundness:** 3 good
**Presentation:** 3 good
**Contribution:** 3 good

**Summary:**

This paper studies adversarial robustness evaluation of defense models. Based on the fact that the accurate robustness evaluation is difficult, this paper proposes a new binarized testing method that can discover weak attacks. Eleven out of thirteen defenses fail the test and could be broken by stronger defenses.

**Questions:**

The proposed binarized test also needs to be designed for different models. How about the efforts?

**Limitations:**

The authors have discussed the limitations and potential negative societal impacts.

**Strengths And Weaknesses:**

This paper studies an important problem and make a great contribution to the field. Although previous studies have been conducted to provide guidelines and practice to develop more accurate robustness evaluation, there truly lacks a formal test whether the robustness is overestimated. This paper proposes a novel testing method, which successfully identifies weak defenses.

The main weakness of this paper is that most adopted defenses have been broken before. Thus it is not surprising to see these defenses cannot pass the test. Evaluating more recent and state-of-the-art defenses can make the paper more convincing. The generalizability of the proposed method for more models is questionable.

---

> ### Author Response · Authors · 2022-08-02
> **Response to Reviewer Rijw**
>
> Dear Reviewer,
>
> Thank you very much for your positive review. We are happy you found our paper to "study an important question and make a great contribution”.
>
> Please find our responses to your points below:
>
> **Comment:** _“most adopted defenses have been broken before”_ \
> **Answer:** You are correct that 11 of the 13 defenses analyzed in this work have been broken before. However, 2 of these defenses were believed to be secure (as they were just recently published) and we are the first to investigate and, consequently, break these defenses. We hope that this eases your concern. Furthermore, we generally understand your concern regarding the generalizability of our proposed test and would like to analyze even more defenses. However, we are not aware of any purely empirical defense (with surprisingly good results) that was released in the weeks/months before the submission deadline. Thus, there is a lack of potential further candidates for our test. If you have any potential candidates in mind, we are happy to consider them in an updated version of our submission.
>
> **Comment:** _“The [...] test also needs to be designed for different models. How about the efforts?”_ \
> **Answer:** The tests presented in our work are applicable to any defense that uses a classifier that can be divided in an (arbitrarily complex) feature encoder and a linear classification readout. As the majority of defenses proposed in the past follows this structure, the effort to apply our test to them is considerably small. For conceptually very different defenses, authors might have to (re-)design the test - however, we expect that in most cases only small modifications to the specific type of defense and not a major redesign will be required.

---

### Official Review · Reviewer_XdpX · 2022-07-11

**Rating:** 7
**Confidence:** 2
**Soundness:** 3 good
**Presentation:** 3 good
**Contribution:** 3 good

**Summary:**

This paper proposes a method to test if an adversarial defense method for deep neural networks is strong enough. By a simple modification of the threat model, the paper introduces a new classifier in which the existence of an adversarial example is guaranteed. Such a modified model can be used to examine whether an attack is strong enough, and hence the evaluation of robustness is convincing. Experimental results show most of the previously published defenses are insufficiently strong.

**Questions:**

In L139:
 >If this is not possible for an original sample xc, we cannot apply the test and, hence, skip the sample

What is the proportion of samples that are skipped in the experiment?

**Limitations:**

The author has discussed the limitation of the paper in Section 5.

**Strengths And Weaknesses:**

Originality & Significance:

The idea of the paper is very interesting and looks novel to me. The results show that most of the previously published defenses are not strong enough under the proposed test, which is a little bit surprising. The paper is therefore of significance in that it could guide future research on this area to consider stronger evaluations.

Quality & Clarity:

The paper is generally well-written. The analogy to refuting proof in complexity theory is interesting. The legend of Figure 1B is not very clear.

---

> ### Author Response · Authors · 2022-08-02
> **Reponse to Reviewer XdpX**
>
> Dear Reviewer,
>
> Thank you very much for your positive review and your valuable feedback! It is very encouraging that you perceived our work as “novel”, “interesting” and “well-written”.
>
> Please find our responses to your points below:
>
> **Comment:** _“The legend of Figure 1B is not very clear.”_ \
> **Answer:** We are sorry that this figure was not as clear as we hoped. We expanded the caption of Figure 1B to make it easier to parse the figure (i.e., we explained that the black box depicts the feasible set of an l-infinity norm-bounded attack).
>
> **Comment:** _“What is the proportion of samples that are skipped in the experiment?”_ \
> **Answer:** Thanks for asking this question. For all defenses except that by Pang et al. 2019 we can successfully set up the binary classifier for all samples. For this specific defense, we notice that the setup fails for approximately half of the samples. We hypothesize that this is due to the stochastic nature of the defense, but will further investigate this. We will add this result and a discussion of it to the revised version of the manuscript.

---

### Author Response · Authors · 2022-08-02
**Summary of Author Responses to Reviewer Comments**

We would like to thank all reviewers for their time and very much appreciate their assessment of our work as a _“very interesting and novel”_ (XdpX), _“insightful”_ (KmTN), and _“well-written”_ (KmTn, XdpX) paper that _“addresses an important issue”_ (Wz6X) and _“makes a great contribution”_ (Rijw). Further, our proposed method was praised for being _“novel”_ (Rijw, Wz6X, KmTn) and _“simple and computationally cheap”_ (Wz6X) with our evaluation being _“strong and clearly demonstrating the effectiveness of the test”_ (KmTn).

We considered the thoughtful suggestions of the reviewers and addressed them in the latest version of the manuscript, which we believe further improved our submission.

Here is a summary of the two main concerns and how we addressed them:

* Missing information for Figure 1 and Algorithm 1: We updated the caption of Figure 1, and the description of Algorithm 1 and 2 to make them clearer and easier to read.
* Test results for adversarially trained models: In a follow-up experiment, we investigated how our test behaves for adversarially trained models and weak attacks.

---

### Meta-Review · Area_Chair_KdGC · 2022-08-22

**Recommendation:** Accept
**Confidence:** Certain

**Metareview:**

This paper proposes a simple yet effective test to identify weak adversarial attacks, and thus weak defense evaluations. Empirical results have revealed insufficiently strong evaluations in 11/13 previous published defenses. To me, the paper studies an important problem and makes a valuable contribution to the active research field of adversatial defense and robustness evalution. I recommend acceptance, and encourage the authors to incorporate the reviewers' comments and suggestions when working on the final version.

**Award:**

No

---

### Decision · Program_Chairs · 2022-09-14

Accept